# The uterine secretome initiates growth of gynecologic tissues in ectopic locations

Jan Sunde [1,2,3]*, Morgan Wasickanin[1], Tiffany A. Katz[2], Laurel Gillette[4], Sanam Bidadi[2], Derek O'Neil[2], Ramya Masand[3,5], Richard O. Burney[1,4,6], Kathleen A. Pennington [2,7]

1 Department of Obstetrics and Gynecology, Madigan Army Medical Center, Tacoma, WA, United States of America, 2 Department of Obstetrics and Gynecology, Division of Gynecologic Oncology, Baylor College of Medicine, Houston, TX, United States of America, 3 Department of Obstetrics and Gynecology, Baylor College of Medicine, Houston TX, United States of America, 4 Department of Clinical Investigation, Madigan Army Medical Center, Tacoma, WA, United States of America, 5 Department of Pathology and Immunology, Baylor College of Medicine, Houston TX, United States of America, 6 Department of Obstetrics and Gynecology, University of Alabama at Birmingham, Birmingham, AL, United States of America, 7 Department of Obstetrics and Gynecology, Basic Sciences Perinatology Research Laboratories, Baylor College of Medicine, Houston, TX, United States of America

* Jan@sunde.cx

**Data Availability Statement:** All relevant data are within the manuscript and its supporting information files.

**Funding:** The author(s) received no specific funding for this work. Some of the authors of this

## Abstract

Endosalpingiosis (ES) and endometriosis (EM) refer to the growth of tubal and endometrial epithelium respectively, outside of their site of origin. We hypothesize that uterine secretome factors drive ectopic growth. To test this, we developed a mouse model of ES and EM using tdTomato (tdT) transgenic fluorescent mice as donors. To block implantation factors, progesterone knockout (PKO) tdT mice were created. Fluorescent lesions were present after oviduct implantation with and without WT endometrium. Implantation was increased (p<0.05) when tdt oviductal tissue was implanted with endometrium compared to oviductal tissue alone. Implantation was reduced (p<0.0005) in animals implanted with minced tdT oviductal tissue with PKO tdT endometrium compared to WT endometrium. Finally, oviductal tissues was incubated with and without a known implantation factor, leukemia inhibitory factor (LIF) prior to and during implantation. LIF promoted lesion implantation. In conclusion, endometrial derived implantation factors, such as LIF, are necessary to initiate ectopic tissue growth. We have developed an animal model of ectopic growth of gynecologic tissues in a WT mouse which will potentially allow for development of new prevention and treatment modalities.

## Introduction

Endosalpingiosis (ES), the growth of uterine tubal tissue ectopically, has been an area of intense study recently following a 2016 retrospective chart review found ES associated with gynecological malignancy in 42% of patients, with a prevalence of approximately 1.5% in 60,000 gynecologic specimens [1]. With intensive pathologic evaluation, we subsequently reported an increased premenopausal prevalence of benign ectopic growths, including ES (37%), endometriosis (EM) (32%), paratubal cysts (47%), and Walthard's nests (ectopic

work are employees of the US Army and Baylor College of Medicine. The views expressed are those of the author(s) and do not reflect the official policy of the Department of the Army, the Department of Defense of the US government or Baylor College of Medicine.

**Competing interests:** The authors have no competing interests to declare.

urothelial cell growths) (29%) in women with at-risk tissues age 31-50, with ES prevalence increasing to 66% after menopause, in contrast to a decrease in EM (5%), while 89% of post-menopausal specimens had some type of ectopic lesion and various lesion types could be found in the same patient [2]. Another recent study reporting EM in 39% of random biopsies of patients with chronic pelvic pain also found much higher prevalence than expected [3].Sampling bias is the likely explanation for retrospective studies reporting an association of these benign peritoneal cavity cellular implants with malignancy [4–6], since the large majority of patients undergoing surgery for benign conditions do not have a histologic evaluation of all tissues as thorough as that performed for malignancy.

A leading theory for EM development is retrograde menstruation, where during menstruation endometrial cells travel via the fallopian tubes into the pelvic cavity via retrograde flow. However, retrograde menstruation occurs in up to 90% of women, while up to 32% of reproductive age women are reported to develop EM rather than the historically reported 6–10% [2,7]. Other factors at play in the development of EM may depend on factors present in the peritoneal fluid, likely migrating from the uterus. A multitude of studies have identified increased levels of a variety of cytokines and growth factors, including, interleukin-1β (IL-1β), IL-6, IL-8, tumor necrosis factor-α (TNF-α), epidermal growth factor (EGF), Fibroblast Growth Factors (FGFs), vascular endothelial growth factor (VEGF), and leukemia inhibitory factor (LIF) in peritoneal fluid of women with EM [8–11]. Interestingly, many of these factors are essential to the process of implantation, including LIF [12].

Therefore, upon considering the ubiquity of benign ectopic growths and their coincidental association with malignant conditions, in addition to the retrograde menstruation theory of EM development [7], the unexplained facts that bilateral tubal ligation and also hysterectomy decrease the risk of ovarian cancer [13–16], and that uterine secretome factors are present in peritoneal fluid of women with ectopic lesions [8–10] we hypothesized that cyclic post-ovulatory uterine secretome factors in the uterine fluid drive free-floating epithelial cells to implant and grow in ectopic locations.

To test this hypothesis, animal models are needed. Normal mouse strains and genetically mutated models of ovarian inclusion cysts (OIC, ES of the ovary) mimic findings we and others report in humans, such as increasing ES incidence with age [4,17–19]. Specified genes have been demonstrated to play a role in the pathogenesis of ovarian inclusion cysts in transgenic models [18,19], but these models may also unexpectedly alter genes normally involved in lesion development and these models are not designed to evaluate the role of uterine secretome factors in ectopic lesion development. Furthermore, while some mouse models of ectopic growth (EM models) have begun to incorporate fluorescent mouse strains to allow for in-vivo visualization of ectopic lesion growth [20], there is a need to optimize these models in order to track lesion progression over time [21]. Therefore, the goals of our current study were two-fold, (1) to refine current mouse models of ectopic tissues to better visualize in-vivo lesion initiation and (2) determine the effects of uterine secretome factors on ectopic lesion growth in our mouse model. To do this we developed a transgenic fluorescent mouse model using tdTomato mouse tissues injected into the peritoneum of WT recipient mice to investigate this "uterine secretome" hypothesis, with the hope of identifying key factors involved in the initial step in the pathogenesis of ectopic growths.

## Materials and methods

### Animals

This study was approved by the Institutional Animal Care and Use Committee (IACUC) at the Madigan Army Medical Center (MAMC) and Baylor College of Medicine (BCM). Care and

procedures for the mice were in accordance with the Helsinki Declaration and Institutional Guide for Laboratory Animals. Mice were housed under a 12/12-hour light/dark cycle at 25 C and 50% humidity and were fed ad libitum with a standard diet and water. Mature 8–12 week old female mice expressing ubiquitous tdTomato (tdT) from the Ai14flox allele (B6.Cg-*Gt (ROSA)26Sortm14(CAG-tdTomato)Hze*/J #:007914, Jackson Labs, Sacramento, CA) were chosen as tissue donors due to their fluorescence and ease of in-vivo imaging tdT mice [18,19,21]. Mature 8–12 week old WT C56BL/6J mice from Jackson Laboratory or the BCM mouse facility were used for donor tissue and as recipients to minimize genetic off-target effects.

## Experimental design

To easily identify ectopic lesions in live mice we developed a mouse model using tdT fluorescent mice as donors (Fig 1) and previously published methods from several prior models of EM in mice that artificially induced menstruation in mice using hormone schedules to induce decidualization and endometrial breakdown [20,22,23]. Once the tdT mouse model was established for optimal lesion visualization, experiments were performed to determine if the uterine secretome plays a role in ectopic lesion development, culminating with experiments evaluating the effect of an exogenous secretome factor, leukemia inhibitory factor (LIF) on ectopic lesion initiation. All experiments were planned with 10–12 mice per arm.

## Synchronization of donor and recipient animals

Donor animals: Menstrual endometrium and subsequently diestrus (secretory) phase endometrium was induced in tdT and WT C57BL/6J donor mice using a hormonal protocol [24] of daily subcutaneous (SC) injections of estrogen in sesame seed oil (1 ul/g of 100 ng/100 μl via 28 G needle) for three days (days 0–2), followed by SC injections of progesterone (0.5 mg) and estrogen (5 ng) in 100 μl in sesame seed oil daily on days 6–8. On day 8 pseudo-decidualization of the uterine horns was performed under anesthesia by inserting the sesame seed oil into the cervix using a device fashioned from a shortened angiocatheter and a syringe. Daily SC

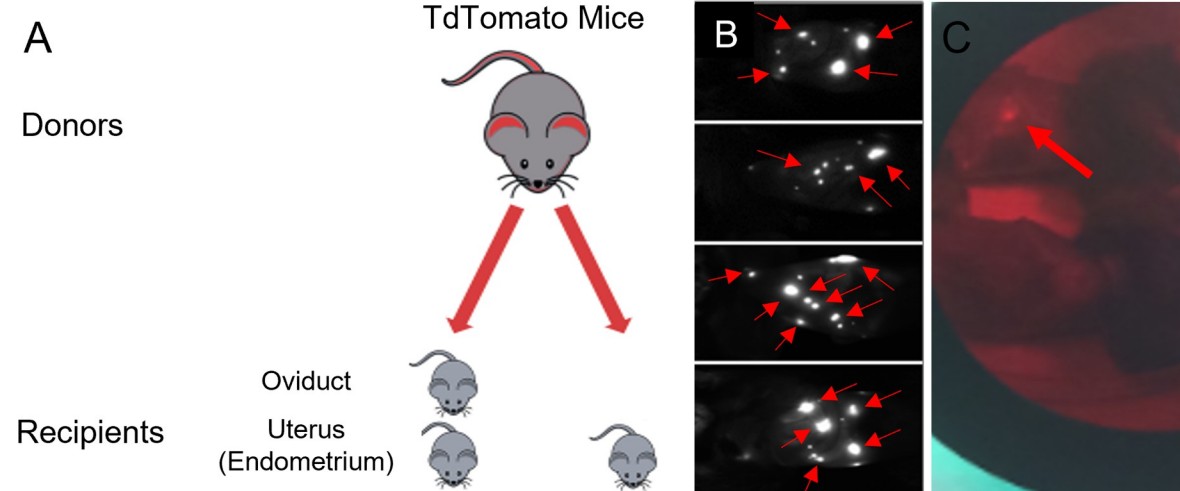

**Fig 1. Development of tdTomato mouse model of endometriosis and endosalpingiosis.** A. After hormonal synchronization, tdT endometrial and oviductal tissue were collected, processed, and injected into WT recipient mice. B. 28 days post injections recipient mice were imaged with a Kodak in-vivo imaging system, visualized fluorescent light from the implanted endometrial or oviductal tdT tissues (red arrows) representing endometriosis and endosalpingiosis mouse models respectively. C. After euthanization, the fluorescent endometrial or endosalpingeal lesions were counted and collected for histological analysis using direct visualization with green light and red filter.

progesterone treatment (0.5 mg) continued for another three days. On day 12, 24–40 hours following the final progesterone injection, mice were euthanized, and tissues were harvested.

Recipient Animals: Female WT C57BL/6 mice were synchronized to diestrus, the cycle phase when secretome factors are expected in the uterine cavity, using progesterone 3ug/200ul saline (15 ug/ml) SC via 28-gauge needle and cloprostenol (prostaglandin F2 alpha), 0.5 ug in 100 ul saline (5 ug/mL) intraperitoneally (IP) via 28-gauge needle followed three days later by a second dose of cloprostenol. This treatment places the recipient mice in the diestrus (post-ovulatory or secretory) phase at the time of tissue injection five days after the initial hormonal injection, which we confirmed histologically. In some studies, male mice were used as recipients in order to eliminate hormonal cycling factors as described below.

## Tissue harvest and implantation

On day 12 of hormone synchronization, donor females were euthanized for harvest of diestrus or menstrual endometrium and/or oviductal epithelium. Post-mortem, the entire uterine horns, and oviducts were removed via sterile technique and placed in a petri dish containing warm sterile PBS media. Under a dissecting microscope, each uterine horn was bisected lengthwise, and the endometrium was dissected from the underlying uterine wall. The isolated endometrium was then suspended in warm media. The endometrium was finely minced with scissors as previously described [25]. Oviductal tissues were obtained by dissection and mincing of the oviducts separately in PBS. PBS and tissues were collected in a micro centrifuge tube and centrifuged for 30–60 sec at 5000 rpm to concentrate the tissue into a pellet and fluid was removed. The harvested tissue/cells and/or minced endometrium from donors was suspended in the supernatant and standardized to a volume 400 μl room temperature PBS before intra-peritoneal injection into WT recipient mice via a 19-gauge needle. The processing time from the dissection of tissue from donor mice to tissue injection in recipient mice was approximately 20–30 minutes.

It was initially anticipated that the ectopic lesion yield for oviductal tissue would be less than endometrial tissue, planning a 4:1 donor: WT mouse ratio using 2 oviducts/mouse on 2 occasions. In a single early iteration, oviductal cells were isolated by flushing the oviduct. With the assistance of a dissecting microscope, oviducts were trimmed of surrounding membrane and straightened before being placed into a fresh petri dish of PBS. Using an instrument to stabilize one end of the oviduct, a 29-gauge syringe was inserted into the end and flushed with 50 μl of PBS, with collection of the flushed fluid for peritoneal injection. Harvesting flushed cells from the oviduct was difficult due to their small size and experimental differences in lesion development were noted using minced tissues, so further experiments were performed with minced tissues to optimize the model. In subsequent experiments, because of the initial extensive lesion development, 1 minced oviduct per recipient mouse (1:2 donor:recipient mouse ratio) was used, with a single peritoneal injection. Minced endometrium from 1 WT diestrus (menstrual phase for initial experiment) uterine horn per recipient mouse was used to provide implantation factors for those mice that were also injected with endometrial tissue (1:2 donor:recipient ratio).

In the initial experiments, donor tissues were injected into the peritoneal cavity of diestrus phase recipient C57BL/6 WT mice, using minced tdT endometrial tissue alone (E), minced tdT oviductal (O) tissue alone, or a combination of tdT oviductal tissue and WT endometrial tissue (EO) from similarly synchronized WT mice donors to provide secretome factors. In all iterations, endometrial, oviductal tissue/cells were harvested from donors and minced/prepared under clean conditions prior to injection into recipient mice. The PBS and oviductal/endometrial tissues were then collected into a micropipette at a volume of 50 μl.

## Optimization and in-vivo evaluation of lesion implantation

WT recipient mice initially underwent live animal (in vivo) imaging under isoflurane anesthesia one week after each minced tissue injection, every 4 weeks with oviductal cells, and prior to post-mortem dissection in the initial experiments to assess for lesion growth prior to harvesting. Critical for the success of the model when using live animal imaging was the optimization of the signal-to-noise ratio in the detection of fluorescent endosalpingiosis lesions. Imaging was performed using a Kodak FX in vivo small animal imaging station (Carestream Health Inc.). Animals were imaged at an excitation wavelength of 555 nm and an emission wavelength of 600 nm for ~30–120 seconds suited for tdT. Imaging allowed for the visualization of lesions and helped provide insight into the progression of the disease and disease burden initially. Depending on the signal-to-noise resolution and lesion size, it was possible for recipient animals to have negative imaging in the presence of small lesions.

Based on live imaging, animals were euthanized and necropsied on day 28 when implanted with minced tissue, with a delay to day 93 when implanted with oviductal cells to assess for implantation of tdT ectopic lesions. At necropsy, lesions were excised from surrounding tissue with the assistance of fluorescence imaging using the NightSea light (Fig 1C) for counting and histology. The tdT endometrium fluoresces well, allowing for visualization of millimeter size lesions. Fluorescent lesions were counted and harvested post-mortem using a red filter with a 550 nm green light for histologic evaluation by H&E staining. Implantation by imaging and at necropsy was assessed by counting the number of lesions, which could be found anywhere in the peritoneal cavity. To minimize abdominal auto-fluorescence from animal feed, the recipient mice were administered a low-fluorescent, alfalfa free purified diet for 15 days prior to any imaging.

## Histology

All lesions and uterine tissue were dissected and fixed in 10% buffered formalin at room temperature for 24 hours. The tissues were processed through ethanol dilutions into paraffin wax. Transverse 5 μm sections were serially cut. To determine gross histology, sections were stained with hematoxylin and eosin (H&E) as per standard protocols (Fig 2C).

Cycle evaluation: We performed an H&E histological cycle evaluation of the endometrium in WT mice after cloprostenol induction and following estrogen and progesterone stimulation on day 12 to confirm that recipient mice were in the secretory phase. Three cloprostanol induction and 1 E/P induction mice were euthanized daily beginning 1 day prior to planned tissue harvest and for 4 additional days for evaluation of the uterine horns.

## Uterine secretome (secretory phase) mouse model

Following development of the tdT mouse model using endometrium, we next sought to evaluate whether tissues other than endometrium can implant ectopically in the model and whether uterine secretome factors may play a role in initial implantation of ectopic tissues. Ectopic tdT oviductal tissue was injected ± synchronized post-ovulatory WT endometrium to provide secretome factors. The first experiment based on prior models, included mouse specific steps including pseudodecidualization by vaginal sesame oil administration in donors to increase the amount of endometrial tissue [20], and administration of SC estrogen 24 hours after tissue injection in recipients based on "nidal estrogen" usage in mouse pregnancy models after ovulation to assist implantation [25–27], as well as tissue injections 2 weeks apart to maximize lesion development. Because numerous ectopic lesions were implanted, the next experiment more closely simulated human secretory cycles by reducing tissue injections to one, eliminating pseudodecidualization, and also estrogen administration. We also tested flushed oviductal cells

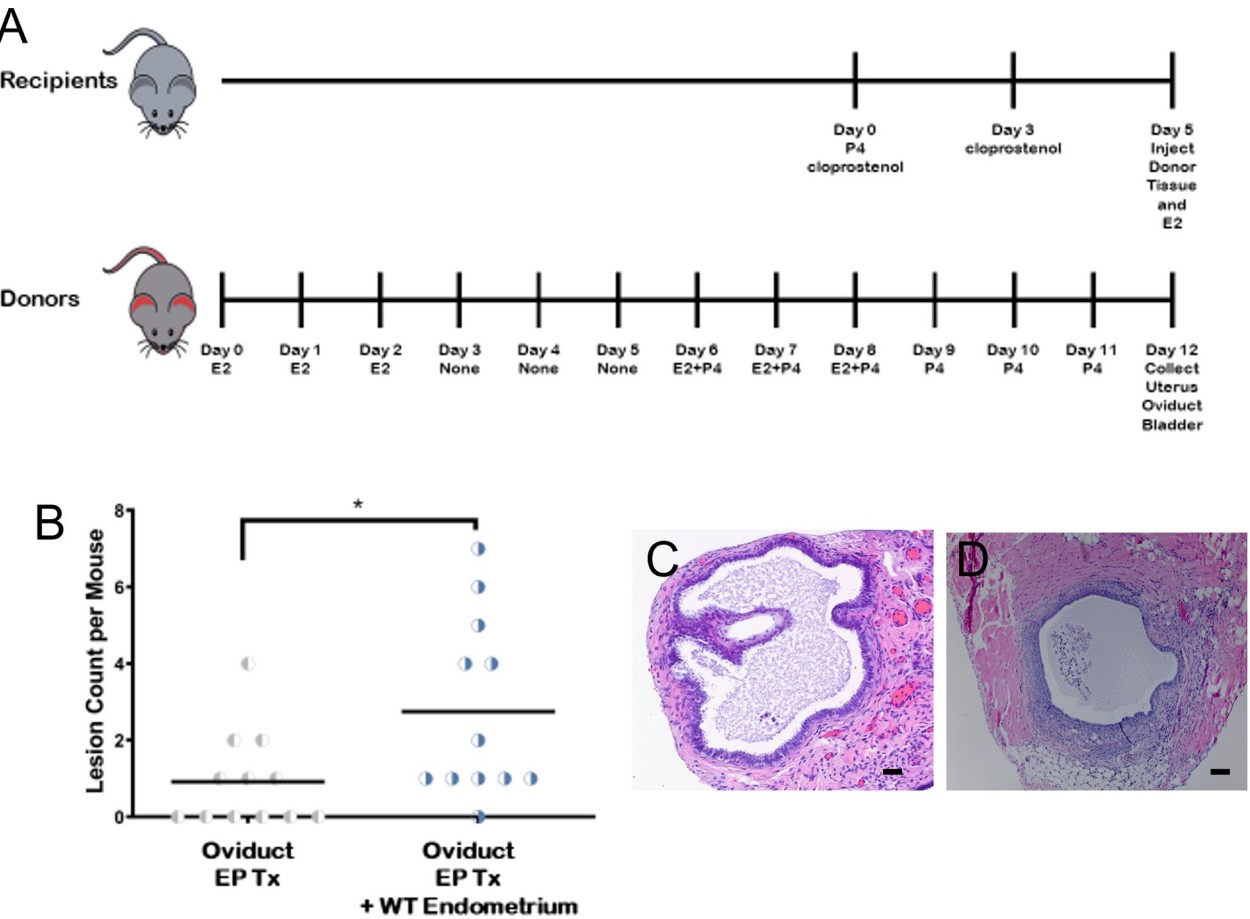

**Fig 2. Co-injection of WT endometrium with oviduct increases ectopic lesions.** A. Timeline for hormonal induction of menstrual/diestrus endometrium for tdT transgenic donors and WT C57BL/6J female recipients. Induction of menstrual/diestrus endometrium. Donor tdT transgenic female mice were hormonally induced with estrogen and progesterone (EP) to develop menstrual/secretory endometrium. WT C57BL/6J recipients were induced using cloprostenol to develop synchronized diestrus endometrium, using experimental and control groups. Endometrial and oviductal tissue were collected, processed, and injected into recipient mice. E2 = estrogen, P4 = progesterone B. The number of implanted oviductal lesions in mice had a statistically significant increase with co-injection of WT endometrium in comparison to solely implanted oviductal lesions. Example histological images of ectopic (C) oviductal and (D) endometrial tissue. Scale bar = 100μm.

to evaluate possible oviductal stromal effect. Subsequent experiments were performed with estrogen and progesterone synchronization to provide adequate diestrus endometrial tissue, since lesions developed in both arms without the eliminated steps. Live and post-euthanasia imaging was performed in these initial experiments at 14 and 28 days, and then monthly for the oviductal cell experiment, but was deemed unnecessary for subsequent experiments once optimal timing of gross lesion evaluation was determined to be 28 days.

### Progesterone Knockout tdT model

To demonstrate the importance of the uterine secretome on ectopic lesions development we superimposed the progesterone-induced uterine gland knockout (PUGKO) model [28] onto our tdT mouse model of ectopic lesions. The PUGKO mouse model results in the ablation of uterine glands and thereby a reduction in uterine secretome factors including LIF [28,29]. tdT mouse pups were injected on postnatal days 2–10 with progesterone 50 ug/g body weight (bw) (5 mg/ml sesame oil) SC with a 26 G needle. The progesterone effect on the endometrium was

evaluated by immunohistochemical staining for uterine gland marker, forkhead box A2 (FOXA2) [28]. Primary antibody recombinant anti-Fox2a antibody (ab108422) and secondary goat anti-rabbit IgG H&L HRP conjugated (ab6721) were obtained from abcam and used according to manufacturer's instructions. DAB Substrate kit (abcam) was used according to manufacturer's instructions for visualization, hematoxylin was used to counterstain nuclei as we previously described [30].

Twelve recipient mice were used in each arm. Normal and PUGKO tdT mice were used as donors. Experimental arms included minced tdT oviduct, minced PUGKO tdT oviduct ± WT endometrium, minced tdT endometrium ± WT endometrium (Fig 3).

## Hormonal effects

Next, we evaluated the effects of hormonal cycling on ectopic lesions development. Cyclicity was assessed in the recipient and donor mice via H&E histology. Synchronized female recipient mice were changed to male mice suppressed with degarelix acetate 0.5 mg SC (5mg/mL sterile water), a GnRH antagonist, 2-weeks before tissue injection to eliminate possible recipient hormonal or other contributions on lesion development, and to decrease mouse wastage. Female donor mice suppressed with degarelix were evaluated to assess the importance of cycling endometrium.

Tissues from synchronized degarelix treated tdT mice and/or endometrial tissue from synchronized WT mice were injected into the peritoneal cavity of degarelix treated male C57BL/6 WT recipient mice, using minced endometrial tissue alone, minced oviductal tissue alone, or a combination of oviductal tissue and minced endometrial tissue from donor tissue incubated with LIF as described below to evaluate the implantation of lesions in both arms.

## Leukemia inhibitory factor supplementation

Finally, the addition of recombinant LIF as a single experimental variable was tested to see if ectopic endosalpingeal and endometrial lesion development increases in its presence.

Oviductal and endometrial epithelium was harvested and injected in a similar fashion ± LIF, using different incubation times and concentration. LIF 10 ug [100 ug/mL] in 100uL was added to 300uL PBS with tissue, incubated for 5 minutes and then injected into recipient mice in the experimental arms. Endometrial tissue with LIF 40 ug in 400 uL PBS with a 30 minutes incubation period was also evaluated.

## Prepubertal Ovarian inclusion cyst (OIC) development

We also evaluated the presence of OICs in pre-pubertal mice to assess whether spontaneous metaplasia might be a possible source of serous lesions. Mice are known to begin ovulating by approximately 35 days, so 100 ovaries of 50 WT C57BL/6 mice euthanized at 30–31 days were assessed for OICs. This number of mice were utilized, since OICs were not expected to be present. The ovaries were fixed in 10% buffered formalin at room temperature for 24 hours. The tissues were processed through ethanol dilutions into paraffin wax. Two transverse 5 μm sections were serially performed in every 20 μm intervals, one slide for H&E staining per standard protocols and the second adjacent slide preserved for IHC staining in case of OIC presence.

## Statistical analysis

Statistical analysis was performed using GraphPad Prism (La Jolla, CA, USA). IPITT were analyzed using 2-way ANOVA with diet and time as factors. For multiple comparisons an ANOVA was used, and Tukey test was used for post-hoc, pairwise comparisons.

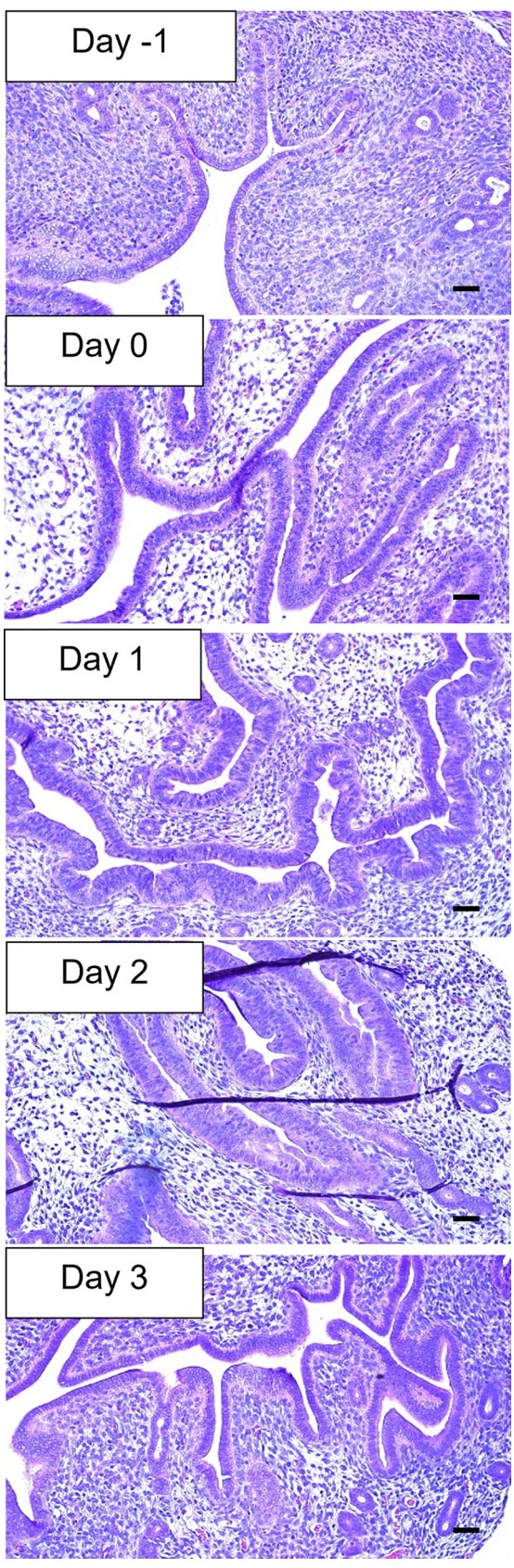

**Fig 3. Cloprostenol treatment induces diestrus in the mouse uterus by day 3 of treatment.** H&E images of the mouse uterus from day -1 to day 3 of cloprostenol treatment. 20x magnification, Scale bar = 50 μm.

## Results

### Model establishment using tdTomato mice

To visualize ectopic lesion growth over time in-vivo, we developed a mouse model using tdT mice as donors and WT mice as recipients. Lesions were visualized 28 days post injection of either endometrial or oviductal donor tissue (Fig 1B). The use of tdT tomato mice as donors also allowed for visualization of lesions at time of necropsy (Fig 1C). All the mice in the initial experiment with two study arms (Fig 1A), one arm receiving minced tdT oviduct alone, and the second receiving minced tdT oviduct and menstrual WT endometrium to provide secretome factors, had extensive implanted lesions, up to 14/mouse. Mice receiving both endometrium and oviduct averaged numerically but not significantly more lesions than those who received oviduct alone (5.8 EO lesions per mouse versus 4.4 O, p = 0.489). Endosalpingiosis/oviductal tissue was implanted in all recipients in both study arms. Overall, the use of tdT mice as donors into WT recipient mice provides in-vivo visualization of lesion development and allows for visualization of small lesions at time of dissection.

### Uterine secretome (secretory phase) factors increase ectopic lesion development

Given that oviductal implantation in the model was successful, and occurred in the absence of supplemental WT endometrium, we proceeded with a series of experiments to determine if the presence of uterine secretome factors play a role in the process of ectopic tissue growth, by eliminating possible confounding variables. Initial factors we considered were: an effect of oviductal stromal cells in the minced oviductal tissue, and two factors distinct in the mouse model compared to humans: uterine horn pseudodecidualization, and the effect of post-injection supplemental estrogen. Initially, all mice were sacrificed on days 21–28 after initial implantation. However, on initial imaging in the experiment using oviductal cells, no lesions were visualized. Due to the concern that individual cells would potentially require a longer time to grow into visible lesions, mice in this experiment were sacrificed 93 days after implantation. The oviductal cell experiment showed lesions in 3/10 mice, 2 in the post injection estrogen arm and 1 in the no estrogen arm (5 with estrogen and 5 without estrogen). Additional experiments were completed using minced tissues due to difficulty harvesting cells alone and the duration of time for lesion growth, since our focus was factors capable of affecting implantation of free-floating tissues. Since lesion development occurred ± pseudodecidualization and also ± post injection estrogen treatment, these treatments were sequentially removed from the experiments to reduce confounders. An experimental timeline is provided (Fig 2A).

Minced tdT oviduct was then implanted ± WT endometrium, with 12 mice/arm (Fig 2B) There was significant increase in the number of oviductal lesions when implanted with endometrium versus without.

An experiment to confirm the effect of cloprostenol treatment to achieve the diestrus phase was performed by H&E evaluation of the endometrium following cloprostanol induction (Fig 3). The histologic evaluation confirmed that mice had secretory endometrium 48 hours after 2 cloprostenol injections (at day 2 following the second cloprostanol injection).

### Ablation of recipient uterine glands significantly decreases implantation of ectopic lesions

Subsequently, we sought to manipulate the uterine secretome and therefore we superimposed the PUGKO uterine receptivity mouse model onto our mouse model of ectopic lesions (Fig 4A & 4B).

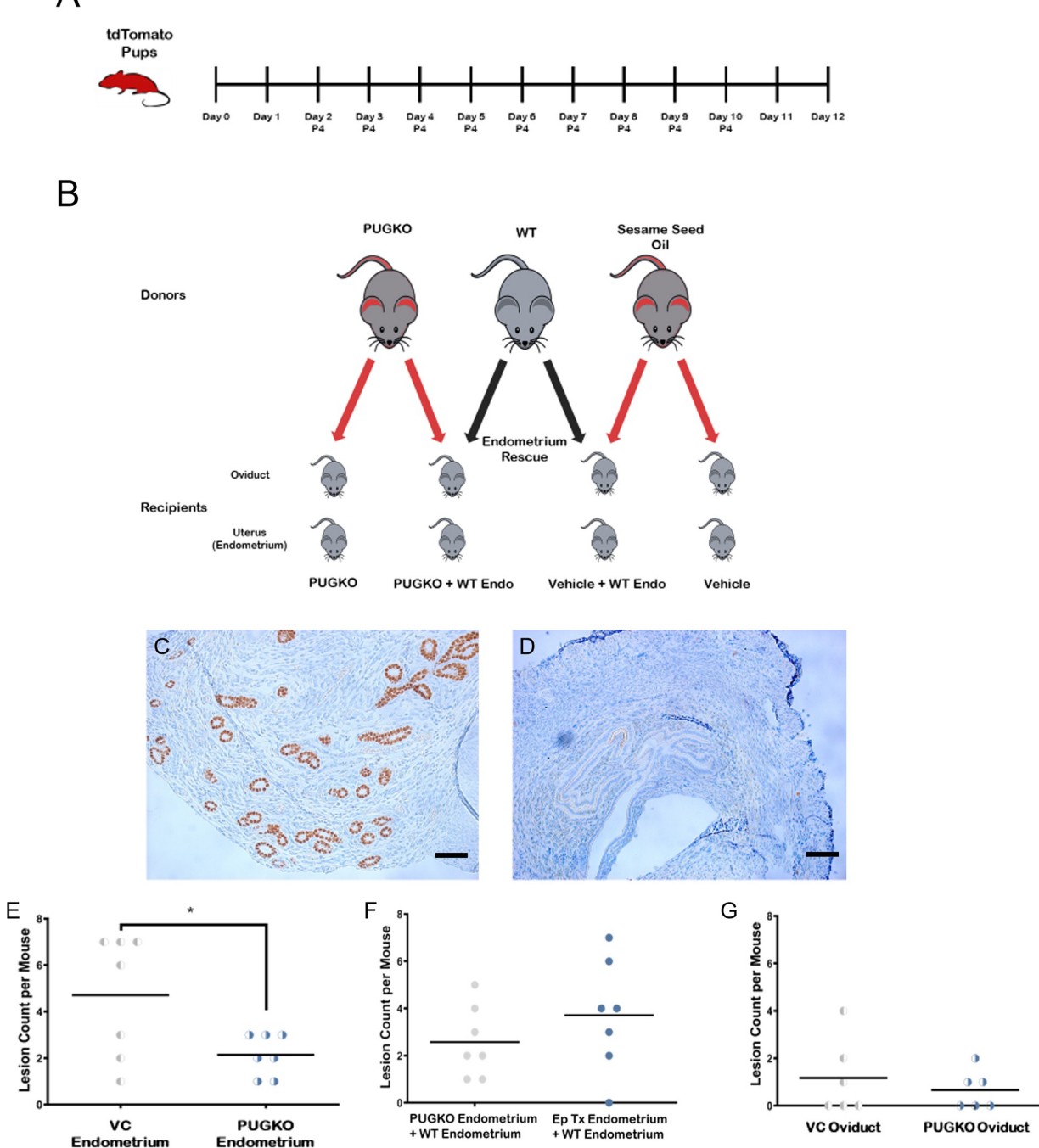

**Fig 4. Ectopic lesions are decreased in PUGKO mice compared to controls.** A. Creation of tdT PUGKO mouse model with daily injections of progesterone from postnatal day 2–10 to knock down secretome factors was based on mouse model that results in infertility in female mice. P4 = progesterone B. Oviduct tissue and endometrium from progesterone treated (PUGKO) and vehicle control donor mice were collected and processed, then injected into WT C57Bl/6J recipient mice. Due to the fluorescent nature of the tdT mouse tissue, all tissue that implanted in the recipient mice could be identified using a red filter with a green light exposure. C -D. Immunohistochemistry (IHC) staining for FOXA2 was performed to assess the response of pups to progesterone treatment (PG TX). This was measured by comparing the counted IHC FOXA2 stained uterine glands of microscopic images in 4 high power fields (HPF) of 40 magnification in PUGKO tdT (D) vs WT (C) Scale Bar = 100 μm. PG treatment in PUGKO mice decreases the expression of FOXA2 almost 33 times compared to the non-PG TX mice. **E.** Lesion count in mice with endometrial tissue from tdT mice ± progesterone on postnatal day 2–10 (PUGKO). There are significantly less lesions when mice recieve progesterone as pups (* = p<0.05). **F.** Lesion development using PUGKO endometrium and secretome factors (provided by synchronized WT endometrium). PUGKO reduces implantation factors and associates with fewer endometrial lesion growths. WT endometrium appears to

restore implantation of PUGKO endometrium. There is no statistically significant difference when PUGKO endometrium is co-injected with WT endometrium. **G.** PUGKO does not significantly alter the number of oviductal lesions. The progesterone induced blocking of uterine glands in PUGKO mice did not statistically affect the number of oviductal lesions compared to controls in vehicle mice, since oviduct does not produce secretome factors. (Student t-test.) E: Estrogen, P: Progesterone.

Uterine gland marker, FOXA2, was reduced in PUGKO compared to control mice (Fig 4C & 4D). Endometrial lesion count was significantly decreased in PUGKO tdT mice compared to tdT mice (p<0.05) (Fig 4E). Lesion development increased with addition of WT endometrium as expected, but this was not statistically significant with the sample size and small lesion numbers (12 lesions/6 mice vs 4 lesions/6 mice p = 0.12) (Fig 4F). Oviductal lesion count was minimally decreased in the tdT PUGKO model compared to tdT tissue, as expected, since FOXA2 is thought to affect uterine secretome factors (Fig 4G).

### Leukemia inhibitory factor and hormonal cycling impact lesion implantation and initiation

Our initial experiments demonstrated that oviductal tissue from tdT mice have increased lesion development in the presence of secretory factors provided by WT endometrium (Fig 2B). In these experiments we wanted to explore whether the role recipient reproductive hormones in the recipient mouse, specifically estrogen and progesterone, affect ectopic lesion development. To do this, we utilized male mice treated with degarelix as recipients. Subsequently, we tested the absence of hormonal stimulation of donor tissue, finding decreased lesion development when donor mice were treated with degarelix to suppress endometrial cycling. Of note, a low baseline implantation rate was noted for tissues that were hormonally suppressed (Fig 5).

Finally, we tested the addition of leukemia inhibitory factor (LIF), a cytokine factor long known to play a role in mouse pregnancy implantation, to endometrial tissue which did not statistically increase the lesion count when given to degarelix suppressed mice with small numbers of mice. (Fig 5A & 5B). On the other hand, hormonally treated endometrium resulted in significantly more lesions than endometrium from mice treated with degarelix, with most lesions in the hormones + LIF injection cohort, demonstrating that hormones are important to prepare the uterine secretome factors that play a role in lesion implantation, and LIF works in combination with other secretome factors (Fig 5C). Oviduct tissue treated with LIF before injection into donor mice also resulted in a significant increase in lesion count (Fig 5D). The addition of the implantation factor LIF increased the implantation of oviductal tissue but not endometrial tissue, presumably because the endometrium secretes endogenous LIF.

### Prepubertal ovarian inclusion cyst assessment

Retrograde flow and implantation as the source of ectopic lesions has been an accepted theory for nearly 100 years and is explained by the secretome theory. An alternate theory is metaplasia of ovarian tissue, so we searched for the presence of OICs in "pre-ovulatory" mice, and found no lesions in these mice that have not yet initiated ovulation (Fig 6).

### Discussion

Our mouse model introduces the "uterine secretome" theory that free floating gynecologic epithelial cells which migrate, most often to the peritoneal cavity, are influenced by uterine secretome factors to implant ectopically. Neither Sampson's retrograde menstruation theory [31], nor the "precursor escape" theory leading to serous ovarian cancer, first proposed in 2003 by

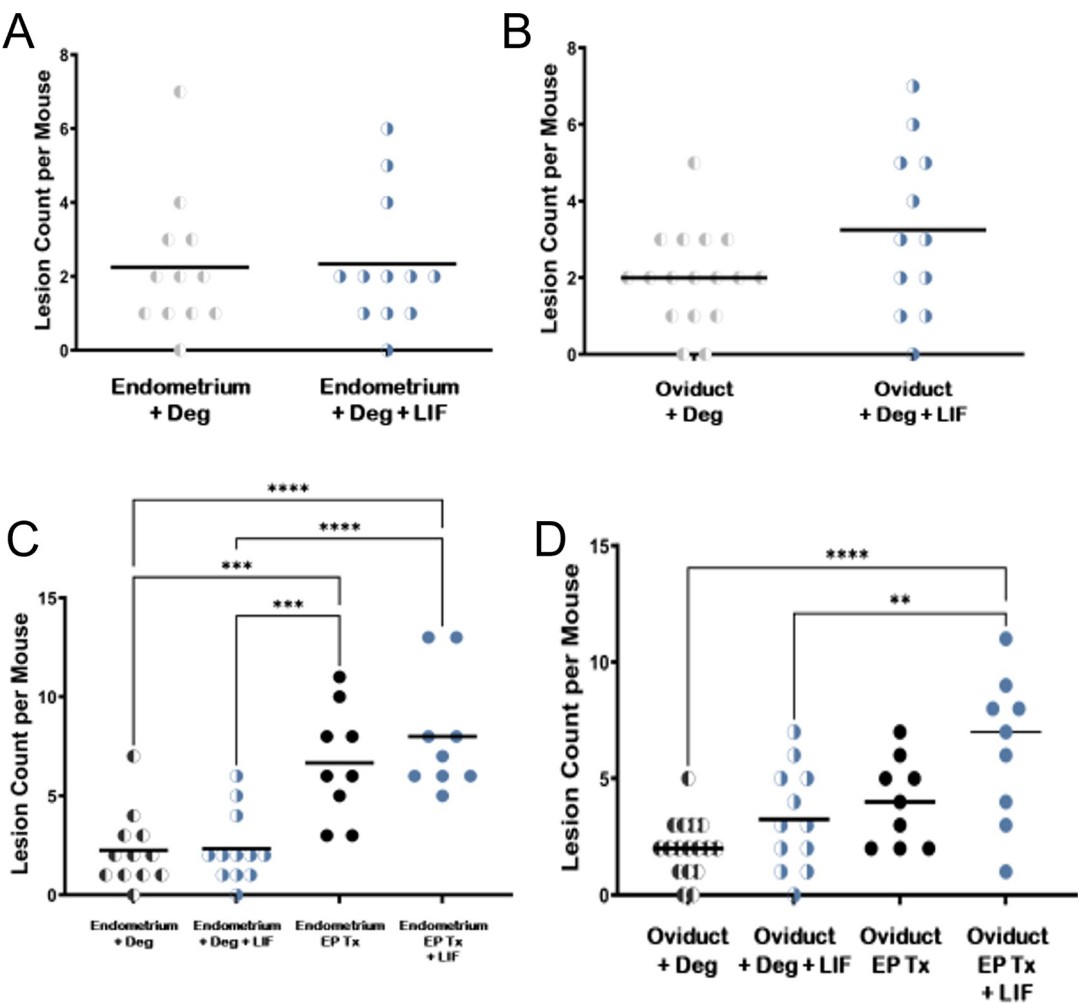

**Fig 5. Effects of LIF on ectopic lesion growth. A.** The addition of LIF does not alter the number of endometrial lesions. When the donor and recipient male mice were hormonally suppressed with degarelix, the co-injection of LIF did not demonstrate any statistically significant change in endometrial lesion/mouse compared to solely implanted endometrial controls. **B.** Addition of LIF alone does not statistically alter the number of oviductal lesions. In hormonally suppressed donor and recipient male mice with degarelix, the co-injection of LIF did not demonstrate significant change in oviductal lesion/mouse compared to solely implanted oviductal controls treated with degarelix. C. Disruption of the estrus cycle by degarelix injection in donor mice significantly reduces the number of endometrial lesions/mouse in comparison with hormonally synchronized controls injected with endometrium (***p<0.001). Addition of exogenous LIF to hormonally synchronized endometrium further significantly increases the number of injected ectopic endometrial lesions/mouse (****P<0.0001). **D** Implantation of oviductal tissue from hormonally synchronized mice with LIF significantly increases the number of lesions/mouse compared to oviductal lesion number in degarelix treated mice ± LIF (**p<0.01) and hormonally synchronized mice without LIF. E:estrogen, P: progesterone.

Piek et al [32] and others [33–35] addresses how ectopic tissues initiate the attachment process, nor do they explain the reported 89% prevalence of benign epithelial ectopic lesions in meno-pausal patients [2]. The uterine secretome theory expands upon the "precursor escape" theory by explaining how benign and malignant lesions are initiated, and also explains the increase of ectopic growths with age, as more secretory cycles lead to more lesions. Distant ectopic growths can be explained by detached endometrial cells, which have now been demonstrated in the bloodstream [36–38] driven by the secretome to implant.

Our tdTomato ectopic lesion model described here, which is based on earlier endometriosis models [20,22,23], successfully implanted many endosalpingial lesions in both arms initially,

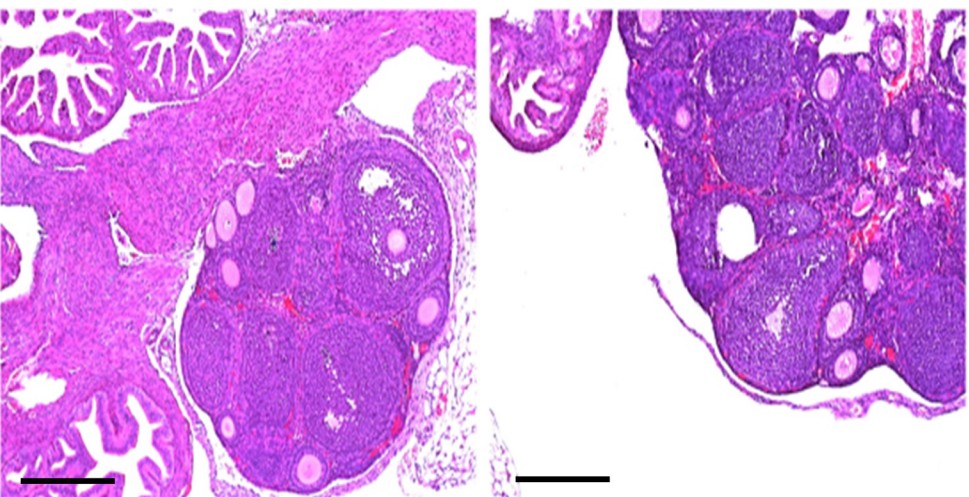

**Fig 6. H&E stained 5 µm sections of two WT C57BL/6 mice ovaries at x10 demonstrate follicles, but no OICs.**
Scale bar = 100 µm.

treating with estrogen and progesterone followed by pseudodecidualization using intrauterine sesame oil to create a menstrual endometrium which is not natural in mice, prior to tissue harvesting, and estrogen supplementation in the recipients ± post-ovulatory WT endometrial tissue to provide secretome factors. Subsequent iterations tested confounding variables in both the donor and recipient mice, beginning with mouse-specific factors. The first variables eliminated were mouse-specific pseudodecidualization associated with uterine stimulation and the estrogen "nidatory surge" which did not prevent ectopic lesion development [22,26]. Minced tissue in the diestrus or "secretory" phase of the cycle achieved by hormonal manipulation was sufficient to increase lesion development, and was easier to work with compared to flushed oviductal cells, so minced tissues were further used to test the effect of the uterine secretome on lesion numbers.

The PUGKO tdT mouse model treated mice with postnatal progesterone on days 2–10 (based on a fertility model that shows decreased embryo implantation as adults) to decrease FOXA2, an upstream regulator of LIF [28,29], which was confirmed by IHC staining of the uterine horn. There is some variability in LIF effect in different mouse species [39] so we did not anticipate elimination of FOXA2. Significantly decreased endometriosis lesion development was found and was restored by supplementation of implanted PUGKO tdT tissues with WT endometrium to provide secretome factors, indicating that uterine factors increase ectopic tissue implantation. Oviductal lesion numbers in PUGKO tdT mice were not significantly different than tdT mice, since the oviduct does not produce secretome factors, but were increased in the presence of WT endometrium (Fig 4).

We demonstrated that there is a baseline level of lesion development even when the donor endometrium is suppressed and that hormonal manipulation to place the endometrium in the secretory phase significantly increases lesion numbers, with a further increase with addition of synchronized WT endometrium to provide secretome factors. Recipient female mice were initially synchronized to the diestrus/secretory phase using cloprostanol induction, and subsequently hormonally suppressed male mice were successfully utilized to eliminate the possibility of recipient uterine secretome factors or other hormonal effects.

To demonstrate the role of a known uterine secretome factor, we supplemented tissue injections with recombinant LIF, which statistically increased oviduct and endometrium

implantation in the presence of hormonally synchronized endometrium. Gene expression studies in embryo implantation [28,29,40], endometriosis [10,11] and ovarian cancer metastasis [41] have all found altered expression of LIF, and other genes such as CD44 and miR 99a-5p [42,43], supporting our hypothesis that the uterine secretome is important in benign and malignant processes. Similar alterations in miRNA expression have been reported in these settings [44,45]. LIF affects the JAK-STAT pathway, which is altered in ovarian metastasis [46]. Endometriosis has been shown to have elevated LIF in the peritoneal fluid [9,10]. An earlier paper reports a 30–40% difference in LIF from uterine flushings of EM patients, which they considered non-significant [47], but the non-significance may be due to small numbers in their study. Additional recent confirmatory data supporting the secretome theory include the association of endometriosis with exosomes that increase invasion and migration [9,47,48], and the persistence of LIF into the menstrual phase in humans [49]. Further study of the overlap of altered gene expression in these processes will be a focus to elucidate the role of these genes and others in the ectopic benign and malignant lesion initiation and metastatic processes.

The "uterine secretome" and "precursor escape" theories are strongly supported by recent data. A 2022 large multi-institutional study showed that risk of primary peritoneal serous carcinoma (PPSC) is increased in women with serous tubal intraepithelial carcinomas (STIC) compared to those with normal Fallopian tube epithelium in women having risk reducing salpingo-oophorectomy (RRSO), with a risk of primary peritoneal serous carcinoma (PPSC) of 10% at 5 years and 25% at 10 years, giving an estimated hazard ratio to develop PPSC during follow-up in women with STIC of 33.9 [50], likely due to implantation of peritoneal pre-malignant lesions prior to RRSO that become malignant over time. Additional studies that support this concept provide data that earlier RRSO provides greater protection [50,51] and that opportunistic salpingectomy decreases multiple ovarian epithelial cancer subtypes [48,52].

It was demonstrated in 1982 that epithelial ovarian cysts typically develop after puberty and that hormonal stimulation is critical to the process [53], likely via the secretome. Our mouse data is consistent with the finding that ectopic gynecologic tissue growth is hormonally driven, as we found no prepubertal OIC lesions in mice.

The pathogenesis of ectopic genetic lesions such as EM and ES has several alternate proposed theories such as coelomic metaplasia & embryonic origin, in addition to retrograde menstruation, with unabated controversy continuing since Sampson first proposed the retrograde menstruation theory [31]. The retrograde menstruation theory as the origin of EM can be subsumed into the uterine secretome theory which finally elucidates the first step in the process after nearly 100 years, and further explains difficulties with the retrograde flow concept. Distant EM & ES, presumably spread hematogenously and via lymphatics, are readily explained, as both endometrial cells and exosomes can be found in women's serum [54] thereby leading to distant lesions originating from the gynecologic tract under the influence of the secretome. Recent research proposing genetic and epigenetic changes as a cause of EM, such as genetic mutations in KRAS, ARID1A, PI3Ka, PIK3ca, ARHGAP35, PP2A in eutopic and ectopic endometrium [55] also support the secretome theory, as the reported genes affect processes driven by the uterine secretome, such as implantation, cell invasion and migration (KRAS [56], ARID1A [57], PI3Ka [58], PIK3ca [59], ARHGAP35 [60], PP2A [61]). These genetic mutations may be the reason why EM, which was reported in approximately 1 / 3 of women aged 31–50 undergoing gynecologic surgery with ovaries removed [2], appears to develop in less than half of the women undergoing retrograde menstruation, which is reported in up to 90% of women [62]. Studies evaluating gene expression in embryo implantation, endometriosis and ovarian cancer [10,40,41,63] have all found altered expression of LIF, and other genetic and epigenetic changes, such as CD44 and miR 99a-5p [42,43], supporting our

hypothesis that the uterine secretome is important in all these processes. miRNA which has been reported in both endometriosis and uterine secretory exosomes, such as miR-302a and let-7b-5p are examples of epigenetic changes [64,65] that will be further investigated. EM has been shown to have elevated LIF in the peritoneal fluid [9,10]. The peritoneal cavity provides an alternate milieu compared to the endometrial cavity, and it is widely reported that the body's immune response mounts an immune response to cells growing ectopically [66]. Just as there are differences in genetic expression between the pre-implantation blastocyst and an implanted embryo, differences in ectopic lesion implantation and subsequent growth are to be expected as we investigate gene expression further.

One alternate theory proposes that ectopic lesions have an embryologic origin, and the presence of EM noted in fetuses is cited in support of this theory [67]. Of note, we found that implanted free floating tissue led to ectopic lesions growth at a low rate, even in the absence of hormonal or secretory stimulation, likely because cells which are not on an apoptosis pathway will look for a nutrient source. This provides an alternate explanation on how retrograde movement of endometrial cells, which are undergoing active proliferation, but not hormonal cycling of the endometrium in the fetus, can lead to the 10% rate of EM [67]. It has been demonstrated that EM lesions decrease dramatically due to the hormonal changes associated with menopause [2], so this would also be expected after birth, when maternal hormones are no longer present. Rare descriptions of both ES and EM found in males [68,69] have been reported as evidence of another theory, coelomic metaplasia, yet these lesions may potentially originate in the remnants of the mullerian structures in males, the prostatic utricle/appendix testis. There is no reported explanation for the lack of ectopic lesions arising in embryologic remnants in males, whereas up to 90% of women have benign ectopic growths once they reach menopause [2]. Our model will allow for experiments to better understand factors that influence the first steps in lesion development.

In conclusion, we developed a mouse model of ectopic lesions that incorporates the ability to visualize ectopic lesions growth in-vivo over time and to identify small implanted lesions readily using tdT transgenic mice as donors. Although mice don't develop endometriosis spontaneously, mouse models have been developed [20–22]. We have refined our model to resemble the human model hormonally closely, because the donor mouse is stimulated to the secretory phase, but not overstimulated to cause menstrual bleeding. We also eliminated the psuedopregnancy and nidatory estrogen steps [26,27]. Advantages over similar natural cycle mouse endometriosis models [23] are that very small lesions are easily identified, and that the very short mouse cycle that changes almost hourly is controlled to be in the secretory phase. Furthermore, the data presented here demonstrates that uterine derived secretome factor/s driven by hormonal cycling are the critical factor/s in initiation of ectopic lesion growth. We also showed that LIF, an essential uterine secretome factor, stimulates ectopic lesion growth. Further studies will utilize our new mouse model to further our understanding of how ectopic lesion growth is initiated in both benign and malignant ectopic lesions.

## Supporting information

**S1 File. Data availability tables for all data presented in the manuscript.**
(XLSX)

## Acknowledgments

We would like to thank Dr. Creighton Edwards for his encouragement and support throughout these studies.

## Author Contributions

**Conceptualization:** Jan Sunde, Richard O. Burney.

**Data curation:** Jan Sunde, Morgan Wasickanin, Tiffany A. Katz, Sanam Bidadi, Derek O'Neil, Kathleen A. Pennington.

**Formal analysis:** Jan Sunde, Derek O'Neil.

**Funding acquisition:** Jan Sunde.

**Investigation:** Jan Sunde, Morgan Wasickanin, Tiffany A. Katz, Laurel Gillette, Sanam Bidadi, Derek O'Neil, Ramya Masand.

**Methodology:** Jan Sunde, Derek O'Neil.

**Project administration:** Jan Sunde.

**Resources:** Jan Sunde.

**Software:** Jan Sunde.

**Supervision:** Jan Sunde, Derek O'Neil.

**Validation:** Jan Sunde, Derek O'Neil.

**Visualization:** Jan Sunde, Sanam Bidadi, Derek O'Neil.

**Writing – original draft:** Jan Sunde, Sanam Bidadi, Derek O'Neil.

**Writing – review & editing:** Jan Sunde, Kathleen A. Pennington.

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
