## [Decision Letter · Decision Letter 0]

10 Oct 2023

PONE-D-23-25727Uterine secretome initiates growth of gynecologic tissues in ectopic locations.PLOS ONE

Dear Dr. Sunde,

Thank you for submitting your manuscript to PLOS ONE. After careful consideration, we feel that it has merit but does not fully meet PLOS ONE’s publication criteria as it currently stands. Therefore, we invite you to submit a revised version of the manuscript that addresses the points raised during the review process.

We look forward to receiving your revised manuscript.

Kind regards,

Antoine Naem, M.D.

Academic Editor

PLOS ONE

Journal Requirements:

3. Please upload a copy of Figure 7, to which you refer in your text on page 26. If the figure is no longer to be included as part of the submission please remove all reference to it within the text.

**Additional Editor Comments:**

Dear Authors,

Your manuscript has been carefully reviewed by experts in the field. According to the review process and my own assessment as an Academic Editor, I believe that your manuscript has merit and suitable for publication in PLoS ONE.

Please address the reviewers comments and revise carefully your manuscript. I would also suggest considering the following book chapter to be discussed and cited in the main text of the manuscript.

https://link.springer.com/chapter/10.1007/978-3-030-90111-0_9

As it was suggested by the reviewers, the pathogenesis of endometriosis should be discussed. As highlighted in the aforementioned chapter, endometriosis has a complex pathogenesis that combined genetic predisposition and epigenetic dysregulations with great influence of the environmental factors and hormonal dysregulations. The endometriotic lesions were found to have cancer-driver mutations despite its benign nature and many miRNAs dysregulations that are related to proliferation and invasion. The important part of understanding the pathogenesis is understanding to which extent animal models of endometriosis are useful and how much do they mimic the disease in humans. Although a definitive answer cannot be given, it would be useful to discuss your results in light of this point.

We are looking forward to receiving your revised manuscript.

With kind regards,

Antoine Naem

Reviewers' comments:

Reviewer's Responses to Questions

**Comments to the Author**

1. Is the manuscript technically sound, and do the data support the conclusions?

Reviewer #1: Yes

Reviewer #2: Yes

Reviewer #3: Yes

2. Has the statistical analysis been performed appropriately and rigorously? 

Reviewer #1: Yes

Reviewer #2: Yes

Reviewer #3: Yes

3. Have the authors made all data underlying the findings in their manuscript fully available?

Reviewer #1: Yes

Reviewer #2: Yes

Reviewer #3: Yes

4. Is the manuscript presented in an intelligible fashion and written in standard English?

Reviewer #1: Yes

Reviewer #2: Yes

Reviewer #3: Yes

5. Review Comments to the Author

Reviewer #1: In this study, Sunde et al., propose a new mouse model to study endometriosis pathogenesis and give some data supporting a causative effect for the uterine secretome. The study is straightforward and might be interesting for the scientific community. I just have some minor comments:

- The English is understandable but needs some improvement in grammar and syntax for clarity.

- Usually, manuscripts submitted for peer review are given with numbered lines; this helps both the work of the reviewers and authors.

- Please add the reference to this sentence: “Specified genes have been demonstrated to play a role in the pathogenesis of ovarian inclusion cysts in transgenic models, but these models may also unexpectedly alter genes normally involved in lesion development and these models are not designed to evaluate the role of uterine secretome factors in ectopic lesion development.”

- In all histological figures, the magnification used, and the reference size are missing, even though some are reported in the Fig legends. Please provide this information on the image.

- In Fig. 1B, please specify to which implantation the images refer to.

- Fig.2A is not mentioned in the text. Please describe Fig.2A before introducing Fig.2B in the text.

- Fig. 3 is very poorly described; please complete it with additional information.

- I suggest fusing the paragraph titled “Hormonal cycling in both donor and recipient mice affect lesion initiation” and the paragraph titled “Leukemia inhibitory factor improves lesion implantation of oviductal tissue” in one.

Reviewer #2: The article is well written and the data are novel and original. I have only few comments

1) The quality of the histological picture shown in figure 3 is poor. I suggest the authors to improve them, because it is difficult for the reader to analyze them

2) Pathogenesis of endometriosis is still not completelty defined and there are other mechanisms proposed more than the Sampson theory and the free floating gynecologic epithelial cells. It would be interesting to discuss the data produced respect to other pathogenetic mechanisms proposed for endometriosis such as celomic metaplasia or embryologic theory

Reviewer #3: Dear Authors

Thank you for your valuable submission. I read your with the utmost interest and pleasure, however the reading was demanding and required a deep immersion in the field.

I found a small letter mistake (mataplasia instead of metaplasia) in the results chapter

and discrepancies in bibliography, where

some additional text was put in: pos 2, 50, 52

full name of the journal instead of the abbreviation: pos 25, 37

Nevertheless, congratulations for your work. It is my pleasure to recommend it for publication.

6. PLOS authors have the option to publish the peer review history of their article (what does this mean?). If published, this will include your full peer review and any attached files.

Reviewer #1: No

Reviewer #2: No

Reviewer #3: No

---

## [Author Response · Author response to Decision Letter 0]

16 Nov 2023

We would like to thank the editor and the reviewers for the comments on our manuscript we have added a more in-depth discussion on endometriosis (lines 497-552). The comments from reviewers are addressed in a point-by-point manner with our responses in red below

Reviewer #1: In this study, Sunde et al., propose a new mouse model to study endometriosis pathogenesis and give some data supporting a causative effect for the uterine secretome. The study is straightforward and might be interesting for the scientific community. I just have some minor comments:- The English is understandable but needs some improvement in grammar and syntax for clarity.

Thank you for this comment we have gone through the manuscript and improved the writing throughout

- Usually, manuscripts submitted for peer review are given with numbered lines; this helps both the work of the reviewers and authors.

Thank you for this comment and we apologize for the oversight on our first submission– line numbers have been added

- Please add the reference to this sentence: “Specified genes have been demonstrated to play a role in the pathogenesis of ovarian inclusion cysts in transgenic models, but these models may also unexpectedly alter genes normally involved in lesion development and these models are not designed to evaluate the role of uterine secretome factors in ectopic lesion development.”

References have been added (line 84)

- In all histological figures, the magnification used, and the reference size are missing, even though some are reported in the Fig legends. Please provide this information on the image.

Thank you for this comment, scale bars have been added to all images 

- In Fig. 1B, please specify to which implantation the images refer to.

Thank you for this comment, arrows have been added to figure 1B to point out the implantation sites we are referring to (line 295)

- Fig.2A is not mentioned in the text. Please describe Fig.2A before introducing Fig.2B in the text.

Thank you for this comment, we have added reference to figure 2A in the results section (line 316)

- Fig. 3 is very poorly described; please complete it with additional information.

Thank you for this comment, both figure 3 as well as the figure 3 legends have been updated (lines 337-338)

- I suggest fusing the paragraph titled “Hormonal cycling in both donor and recipient mice affect lesion initiation” and the paragraph titled “Leukemia inhibitory factor improves lesion implantation of oviductal tissue” in one.

Thank you for this comment, we have combined these sections under the new heading of “Leukemia inhibitory factor and hormonal cycling impact lesion implantation and initiation” line 377

Reviewer #2: The article is well written and the data are novel and original. I have only few comments1) The quality of the histological picture shown in figure 3 is poor. I suggest the authors to improve them, because it is difficult for the reader to analyze them

Thank you for this comment – we have updated figure 3 to include more clear and larger images

2) Pathogenesis of endometriosis is still not completelty defined and there are other mechanisms proposed more than the Sampson theory and the free floating gynecologic epithelial cells. It would be interesting to discuss the data produced respect to other pathogenetic mechanisms proposed for endometriosis such as celomic metaplasia or embryologic theory

Thank you for this comment, we have added more discussion on endometriosis lines 499-554

Reviewer #3: Dear AuthorsThank you for your valuable submission. I read your with the utmost interest and pleasure, however the reading was demanding and required a deep immersion in the field.

I found a small letter mistake (mataplasia instead of metaplasia) in the results chapter

Thank you for this comment this has been fixed (line 422)

and discrepancies in bibliography, wheresome additional text was put in: pos 2, 50, 52full name of the journal instead of the abbreviation: pos 25, 37

Thank you for these comments – we have fixed these citation errorsNevertheless, congratulations for your work. It is my pleasure to recommend it for publication.

Thank you for your kind comments

Journal Requirements:

1. Please ensure that your manuscript meets PLOS ONE's style requirements, including those for file naming. The PLOS ONE style templates can be found athttps://journals.plos.org/plosone/s/file?id=wjVg/PLOSOne_formatting_sample_main_body.pdf andhttps://journals.plos.org/plosone/s/file?id=ba62/PLOSOne_formatting_sample_title_authors_affiliations.pdf

We have verified the formatting meets PLOS ONE’s style requirements

There is no data to deposit in a data repository, all raw data are provided in the data repository tables 

3. Please upload a copy of Figure 7, to which you refer in your text on page 26. If the figure is no longer to be included as part of the submission please remove all reference to it within the text.

Thank you for this comment, this figure reference should have read figure 4 – it has been corrected

A caption for the supporting data availability files has been added

We have reviewed all our references for correctness and to make sure none of them have been retracted

Acknowledgment changed to comply with guidelines.

---

## [Decision Letter · Decision Letter 1]

19 Dec 2023

PONE-D-23-25727R1The uterine secretome initiates growth of gynecologic tissues in ectopic locations.PLOS ONE

Dear Dr. Sunde,

Thank you for submitting your manuscript to PLOS ONE. After careful consideration, we feel that it has merit but does not fully meet PLOS ONE’s publication criteria as it currently stands. Therefore, we invite you to submit a revised version of the manuscript that addresses the points raised during the review process.

We look forward to receiving your revised manuscript.

Kind regards,

Antoine Naem, M.D.

Academic Editor

PLOS ONE

Journal Requirements:

Additional Editor Comments:

Dear Authors,

Please address the reviewer comment regarding your revised manuscript. In addition, the explanation of the endometriosis pathogenesis requires more details and more recent references. I highly recommend elaborating on the multifactorial pathogenesis of endometriosis and stressing on the importance of the epigenetic factors (such as miRNAs dysregulation) in controlling the endometriosis behavior. For this specific reason, any animal model is not a perfect match for the human endometriotic lesions due to important role of the genetic, epigenetic, and environmental factors that influence the endometriosis progression. Please discuss and cite the following chapter: https://link.springer.com/chapter/10.1007/978-3-030-90111-0_9

Reviewers' comments:

Reviewer's Responses to Questions

**Comments to the Author**

1. If the authors have adequately addressed your comments raised in a previous round of review and you feel that this manuscript is now acceptable for publication, you may indicate that here to bypass the “Comments to the Author” section, enter your conflict of interest statement in the “Confidential to Editor” section, and submit your "Accept" recommendation.

Reviewer #1: All comments have been addressed

Reviewer #2: All comments have been addressed

Reviewer #3: (No Response)

2. Is the manuscript technically sound, and do the data support the conclusions?

Reviewer #1: Yes

Reviewer #2: Yes

Reviewer #3: Yes

3. Has the statistical analysis been performed appropriately and rigorously? 

Reviewer #1: Yes

Reviewer #2: Yes

Reviewer #3: Yes

4. Have the authors made all data underlying the findings in their manuscript fully available?

Reviewer #1: Yes

Reviewer #2: Yes

Reviewer #3: Yes

5. Is the manuscript presented in an intelligible fashion and written in standard English?

Reviewer #1: Yes

Reviewer #2: Yes

Reviewer #3: Yes

6. Review Comments to the Author

Reviewer #1: (No Response)

Reviewer #2: (No Response)

Reviewer #3: Dear Authors

Below you can find the excerpt from your answer

"and discrepancies in bibliography, where

some additional text was put in: pos 2, 50, 52

full name of the journal instead of the abbreviation: pos 25, 37

Thank you for these comments – we have fixed these citation errors"

In my copy there is no change in positions mentioned above.

Please doublecheck it and send the correct version.

7. PLOS authors have the option to publish the peer review history of their article (what does this mean?). If published, this will include your full peer review and any attached files.

Reviewer #1: No

Reviewer #2: **Yes: **The authors have addressed all the reviewers' comments

Reviewer #3: No

---

## [Author Response · Author response to Decision Letter 1]

21 Dec 2023

We apologize for the failure of the citations to update properly, these changes had been made however our reference manager software added them back which was an oversight on our part. We have now corrected and confirmed these changes

The updated citations are as follows in the final version (line numbers from the tracked changes version):

579 2. Sunde J, Wasickanin M, Katz TA, Wickersham EL, Steed DOE, Simper N. Prevalence of endosalpingiosis and other benign gynecologic lesions. PloS one. 2020;15(5):e0232487. doi: 10.1371/journal.pone.0232487. PubMed PMID: 32401810

655 25. Somigliana E, Vigano P, Zingrillo B, Ranieri S, Filardo P, Candiani M, et al. Induction of endometriosis in the mouse inhibits spleen leukocyte function. Acta obstetric et gynecolog Scand. 2001;80(3):200-5. doi: 10.1034/j.1600-0412.2001.080003200.x. PubMed PMID: 11207484.

690 37. Grasso A, Navarro R, Balaguer N, Moreno I, Alama P, Jimenez J, et al. Endometrial Liquid Biopsy Provides a miRNA Roadmap of the Secretory Phase of the Human Endometrium. J clin endocrinolmetab. 2020;105(3). doi: 10.1210/clinem/dgz146. PubMed PMID: 31665361.

734 50. Steenbeek MP, van Bommel MHD, Bulten J, Hulsmann JA, Bogaerts J, Garcia C, et al. Risk of Peritoneal Carcinomatosis After Risk-Reducing Salpingo-Oophorectomy: A Systematic Review and Individual Patient Data Meta-Analysis. J Clin Oncol. 2022;40(17):1879-91. doi: 10.1200/JCO.21.02016. PubMed PMID: 35302882

745 52. Hanley GE, Pearce CL, Talhouk A, Kwon JS, Finlayson SJ, McAlpine JN, et al. Outcomes From Opportunistic Salpingectomy for Ovarian Cancer Prevention. JAMA Netw Open. 2022;5(2):e2147343. doi: 10.1001/jamanetworkopen.2021.47343. PubMed PMID: 35138400

Thank you again for the opportunity to share our work.

---

## [Editor Report · Decision Letter 2]

15 Jan 2024

The uterine secretome initiates growth of gynecologic tissues in ectopic locations.

PONE-D-23-25727R2

Dear Dr. Sunde,

We’re pleased to inform you that your manuscript has been judged scientifically suitable for publication and will be formally accepted for publication once it meets all outstanding technical requirements.

Kind regards,

Antoine Naem, M.D.

Academic Editor

PLOS ONE
---

## [Editor Report · Acceptance letter]

26 Apr 2024

PONE-D-23-25727R2 

PLOS ONE

Dear Dr. Sunde, 

I'm pleased to inform you that your manuscript has been deemed suitable for publication in PLOS ONE. Congratulations! Your manuscript is now being handed over to our production team.

Kind regards, 

on behalf of

Dr. Antoine Naem 

Academic Editor

PLOS ONE